

# Microbial diversity and abundance in loamy sandy soil under renaturalization of former arable land

Audrius Kacergius and  Diana Sivojiene

Voke branch of Institute of Agriculture, Lithuanian Research Centre for Agriculture and Forestry, Vilnius, Lithuania

## ABSTRACT

The abundance and taxonomic diversity of different physiological groups of bacteria and fungi and yeasts in the fields of the long-term experiment of renaturalization of infertile arable soils were studied. The experiment involved four land conversion methods: conversion of arable land to cultivated meadow, soil and forest, leaving the experimental area of arable land. With these studies, we have begun to fill research gaps related to the taxonomic and functional diversity of soil microorganisms. The greatest changes in the abundance of cultivable organotrophic, diazotrophic and nitrifying bacteria were found to be observed in those areas where anthropogenic activities took place, i.e. in the cultivated field and in the cultural grassland. The abundance of bacteria was relatively lower and that of fungi was higher in the soil and in the cultivated area. It was also found that the higher jumps in the abundance of diazotrophs and nitrifiers during the respective stages of vegetation were caused by the applied agrotechnical measures and the cultivation of the respective plants. The abundance of cultivable bacteria was up to $10^5$, and the number of fungi was $10^3$ CFU in 1 g of dry soil. The taxonomic structure was determined by Next Generation Sequencing. The taxonomic groups of *Actino-* and *Proteobacteria* had the highest abundance. The highest number of fungal OTU was distinguished by *Ascomycota* fungi (37–42% of the total number of fungi). Comparing the taxonomic structure of all studied samples, the area planted with pines stands out, where an increase in the taxonomic group of *Basidiomycota* fungi (up to 24%) is observed at the expense of *Ascomycota* fungi. In order to have a balanced, fully rich soil, efforts must be made to maintain a stable structure of microbial communities, which can only be achieved through targeted research.

## INTRODUCTION

Farming on low-productivity soils, traditional agricultural activities are often unprofitable, and the establishment of new forests and grasslands can be one of the most efficient ways of using these lands to keep them unused. Such renaturalizations increase the biodiversity of ecosystems through low-intensity agriculture and afforestation, reduce gas emissions and can therefore be seen as positive factors from an environmental point of view (*Callesen & Ostergaard, 2008*; *Armolaitis et al., 2013*). Over the past 50 years, afforestation of abandoned land, usually completely empty, has become more common—especially in the United States

Corresponding author
Audrius Kacergius,
audrius.kacergius@inbox.lt,
Audrius.Kacergius@lammc.lt

and the United Kingdom. Meadows and pastures across Europe are currently being turned back into forests. China, India and the countries of North and Central Africa, the Middle East and Australia are implementing afforestation projects (*Chen et al., 2019*; *Freer-Smith et al., 2019*; *IUCN, 2020*). Renaturalization processes are currently taking place quite rapidly in some parts of Lithuania and this trend is likely to continue in the future. It is predicted that with the development of non-agricultural activities, forested areas will gradually establish themselves in place of the agrarian landscape that has prevailed for many centuries. However, there is a lack of detailed research on this topic.

Summarizing the data of the first research decade, and the results of subsequent years (*Kazlauskaite-Jadzeviče et al., 2019*; *Tripolskaja et al., 2022*), the transformation of field crop rotation soils into various phytocenoses can be described as a complex of factors with a significant effect on the accumulation of energy and nutrients, in which soil microorganisms play an important role. Adding some components and/or suppressing other existing components can be expected to achieve or improve the desired result. Therefore, in this regard, it is important to know the composition of these microorganism communities, as well as the abundance of individual taxonomic groups. In both the soil rhizosphere and the rhizoplan, bacteria and fungi interact closely with each other. Bacteria also play a very important role in promoting plant growth by increasing the nutrients available to plants, producing phytohormones, and inhibiting the development of soil pathogens (*Ahemad & Kibret, 2014*; *Backer et al., 2018*). In addition, the population structure of microorganisms changes in space and time and is affected by the availability of C, N resources, diurnal t°, porosity, moisture electrolyte concentration, pH changes and oxygen availability (*Girvan et al., 2003*).

The intensity of microbial activity is not necessarily related to their taxonomic diversity, as biogeochemical processes are determined by the activity of active microorganisms. However, despite the importance of active microbes, most research methods are designed to estimate total microbial biomass without estimating its active fraction. Active microorganisms account for about 0.1–0.2% of the total microbial biomass and very rarely exceed 5% in soils without readily available substrates. Potentially active microorganisms, ready to absorb available substrates within a few hours, account for 10 to 40%, and sometimes up to 60% of the total microbial biomass. The number of microbes at dormant state, depending on the agroeco-biochemical characteristics of the soil, can be from 42 to 66% of the total microbial biomass. The transition from a potentially active state to an active one occurs in a few minutes, but the transition from a dormant state to an active state can take from several hours to several days (*Maraha, Backman & Jansson, 2004*; *Barra Caracciolo et al., 2009*; *Busse et al., 2009*; *Blagodatskaya & Kuzyakov, 2013*). One of the simpler methods, plate-count techniques, allows the assessment of most active/potentially active microorganism groups by functional-trophic specialization using selective nutrient media (*Néble et al., 2007*; *Sanchez-Peinado et al., 2009*).

Soil microorganisms in various Lithuanian soils have already been studied to some extent, but only in a fragmented way and without delving into taxonomic diversity (*Piaulokaite-Motuziene, Končius & Lapinskas, 2005*; *Piaulokaite-Motuziene & Končius, 2006*; *Bakšiene et al., 2007*; *Bakšiene et al., 2009*; *Janusauskaite, Ozeraitiene & Fullen, 2009*; *Bakšiene et al.,*

*2014*). The main research gaps of all these studies are related to the absence of detailed studies of both taxonomic and functional diversity of microbes. There is also a lack of clarification of the dependence of the structure of soil communities on the agrochemical properties of the soil. With detailed information of this kind, additional measures can be envisaged to help speed up the renaturalization of soils. Preparations of mycorrhizal fungi are already often used in the case of afforestation. Since there are no detailed microbiological studies of the soil in our region, we started to analyze the soil microorganism communities comprehensively, *i.e.,* determining their structure and composition. We hypothesized that renaturalization of former arable soil will change the abundance and diversity of microbes that may determine soil agrochemical properties, and that full use of information from soil microbial communities can improve soil productivity, maintain prehistory and sustainability. The aim of the research is to determine the qualitative and quantitative parameters of low-performing agro-ecosystem soil microorganism groups caused by different land use systems.

## MATERIALS & METHODS

### Study site and soil sampling

The study area ($\sim$54°34′N, 25°05′E) is situated in East part of Lithuania, East Europe, in the northern part of the temperate climate zone (Fig. 1). The study was conducted as a part of long-term experiment, started in 1995. The experiment was arranged as land-use change of former arable field into fertilized/unfertilized managed grassland (MGf and MGu), soiled field (SA), Pine afforested field (PA), and left cropland field (fertilized/non-fertilized (Cf and Cu)). During the long-term experiment, various biological and agroecological properties was analyzed separately (*Tripolskaja et al., 2022*), excluding soil microorganisms. Soil samples for microbiological analysis were collected as previously described in *Sivojiene et al. (2021)*.

### Quantification of cultivable bacteria and fungi

Cultivable microbial quantification was performed by plate-count techniques using different selective media: Meat Peptone Agar (MPA) (Liofilchem, Italy) for organotrophic bacteria, Starch Ammonia Agar (SAA) for bacteria using the mineral source of nitrogen (*Kuster, 1959*), Ashby's Mannitol Agar—for nonsymbiotic diazotrophic bacteria (*Aquilanti, Favilli & Clemeti, 2004*), and Sabouraud CAF agar (Liofilchem, Italy)—for filamentous fungi and yeasts/yeast-like fungi. The number of bacterial and fungal colony forming units (CFU) was calculated per gram of dry soil (*Carter & Gregorich, 2007*).

### Soil DNA extraction and microbiomic analysis

Pooled soil samples for metagenomic analysis were taken from topsoil layer 10–20 cm depth in summer 2020. Total genomic soil DNA from six soil samples was extracted using the ZR Soil Microbe DNA MiniPrepTM (50) (Zymo Research, Irvine, CA, USA) DNA extraction kit according to the manufacturer's instructions. NGS analysis was performed with BaseClear BV (Leiden, the Netherlands) service using the Illumina NovaSeq 6000 or MiSeq system. The sequences generated with the MiSeq system were performed under

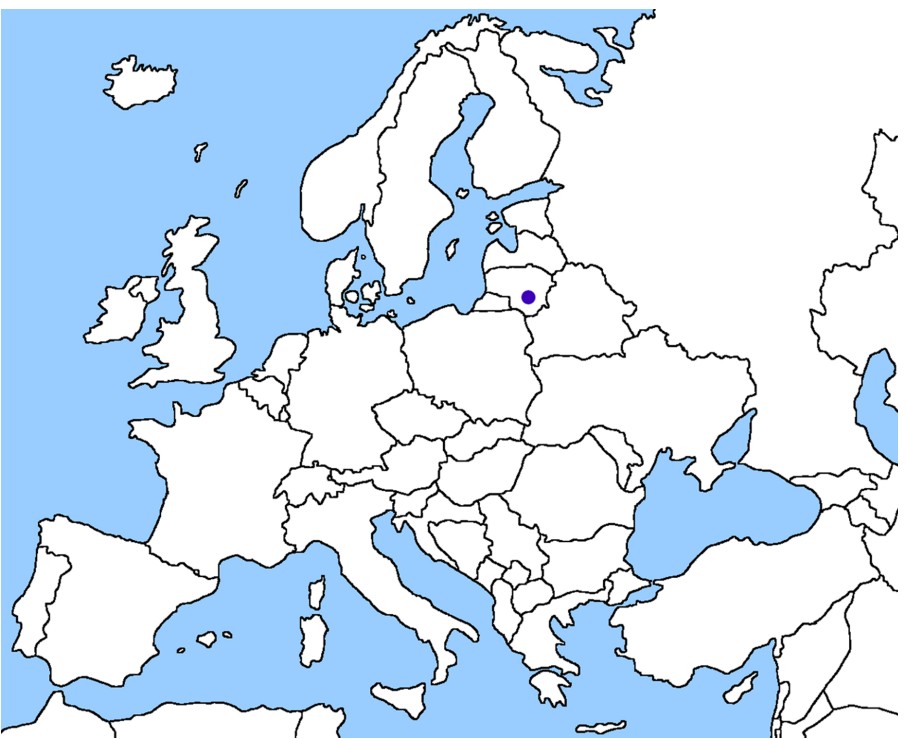

**Figure 1 Geographical location of sampling site.** Map adapted from: https://en.m.wikipedia.org/wiki/File: Europe_blank_map.png.

accreditation according to the scope of BaseClear B.V. (L457; NEN-EN-ISO/IEC 17025) based on 16S rDNA for bacteria and 5.8S-ITS2 for fungi. Paired-end sequence reads were collapsed into so-called pseudoreads using sequence overlap with USEARCH version 9.2 (*Edgar, 2010*). Classification of these pseudoreads is performed based on the results of alignment with SNAP version 1.0.23 (*Zaharia et al., 2011*) against the RDP database (*Cole et al., 2014*) for bacterial organisms, while fungal organisms are classified using the UNITE ITS gene database (*Abarenkov et al., 2010*).

## Climate conditions

Lithuania is in the Northern part of the temperate climate zone. The meteorological conditions of the research years were strictly different. The average temperature in 2017 was close to the multi-annual average, but it was very wet throughout the year. Meanwhile, 2018 was dry and warm, and 2019-2020 was the hottest in the entire almost 240-year (1778–2020) observation period, and there was a significant lack of moisture. According to LHMT, in 2020 it surpassed the warmest ones until 2019, when the average annual air temperature of 8.8 °C was registered and in 2015 (8.3 °C). Annual precipitation in 2020 was 646 mm, which is only 7% less than the multi-annual rate (694 mm) (http://www.meteo.lt/en). Graphic images of meteorological conditions are shown in Fig. 2, were comparison with multi-annual rate (1991–2020) was used.

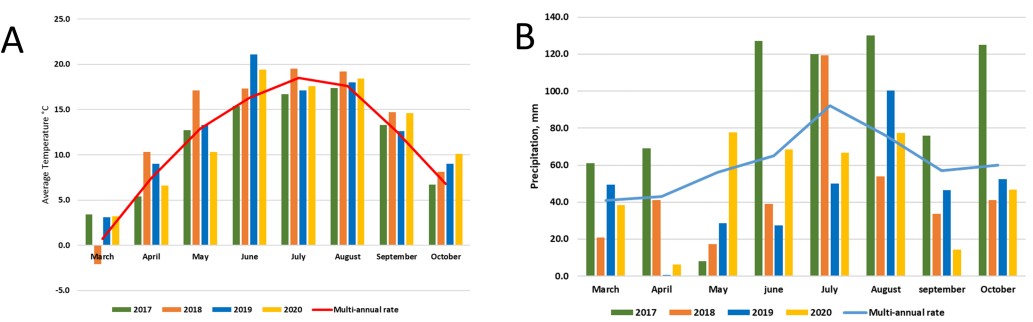

**Figure 2** Meteorological conditions: temperature (A) and precipitation (B) during experimental period (2017–2020).

## Statistical analysis

Microbial abundance data reported as mean ± standard error of the mean and were analyzed using ANOVA. Mean separations were made for significant effect with $F$-test at $0.0000 < p < 0.022$. Taxonomic diversity of microbes was assessed based on the amount of OTUs. Alpha diversity metrics (Chao1 and Shannon) were used to express soil microbial community structure. The Chao1 index describes the abundance of species, while the Shannon index—the diversity of species in given community. Statistical computations were performed using the STATISTICA 16.0 software package (StatSoft, Inc. Tulsa, OK, USA).

## RESULTS

### Soil agrochemical features

During the long research period (23 years), the agrochemical indicators have changed slightly. Soil pH changed the mostly in the unfertilized cropland (increased) and in the fertilized cropland (decreased), while the concentration of organic carbon decreased the most in the unfertilized crop rotation field (*Tripolskaja et al., 2022*).

### Abundance of cultivable soil microorganisms

In 2017, the highest abundance of diazotrophic bacteria in the summer period was observed in the cropland and in the cultivated grassland MG (Fig. 3). Fungi and yeasts were characterized by a high abundance compared to the next year during the summer and autumn periods of this year (Fig. 4). In 2018, diazotrophs and nitrifiers were again more abundant than in other groups in the cropland and in the cultivated grassland (Fig. 3). This may be related to the crops being grown and their fertilization. Barley with red clover undersowing and fertilized with $N_{60}P_{60}K_{100}$ was currently grown in the cropland. In 2019, an increase in the physiological groups of some bacteria was observed in the autumn in a cropland where red clover was grown without fertilization. Increasing amounts of diazotrophs and nitrifiers from the summer period towards autumn were also found here (Fig. 5). The amounts of fungi and yeasts were exceptionally higher in the fertilized part of the cultivated grassland during the summer and autumn (Fig. 4). In the spring period of 2020, significantly higher amounts (not statistically different between themselves) of

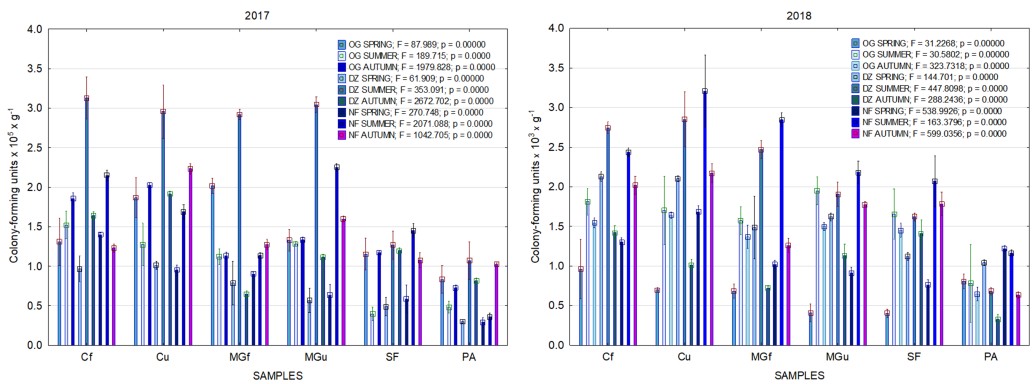

**Figure 3** The abundance of cultivable bacteria in samples of renaturalized area in 2017–2018 (OG—organotrophic, DZ—diazotrophic and NF—nitrifiers).

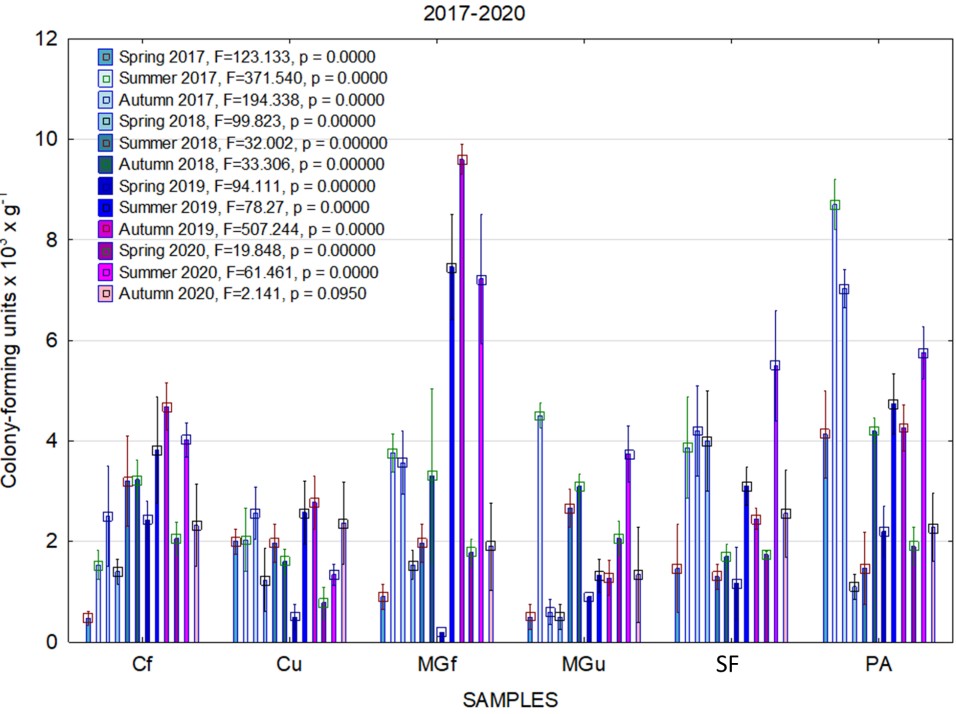

**Figure 4** The abundance of cultivable fungi in samples of renaturalized area in 2017–2020.

diazotrophs and nitrifiers in the fertilized areas of the cropland and cultivated grassland were detected (Fig. 5).

## Soil microbiomic analysis

The method of next generation sequencing (NGS) of molecular biology was used to determine the taxonomic composition of bacteria and microscopic fungi in summer 2020 soil samples. A total of 295,390 valid reads of the 16S RNA fragment of bacteria were

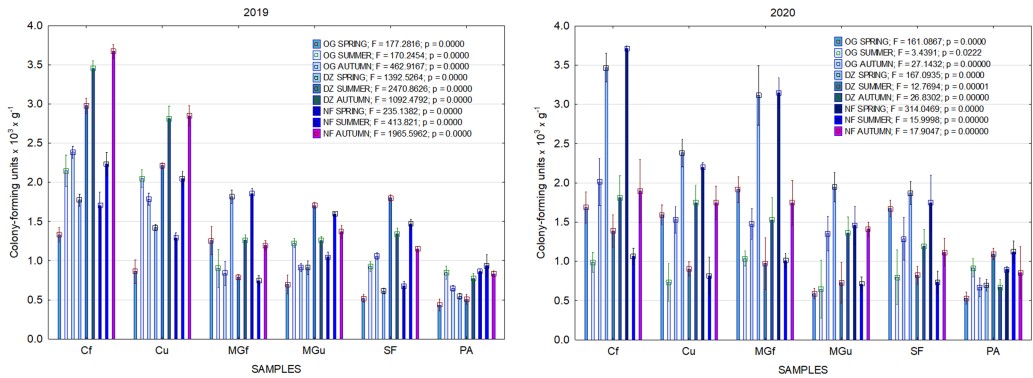

**Figure 5** The abundance of cultivable bacteria in samples of renaturalized area in 2019–2020 (OG—organotrophic, DZ—diazotrophic and NF—nitrifiers).

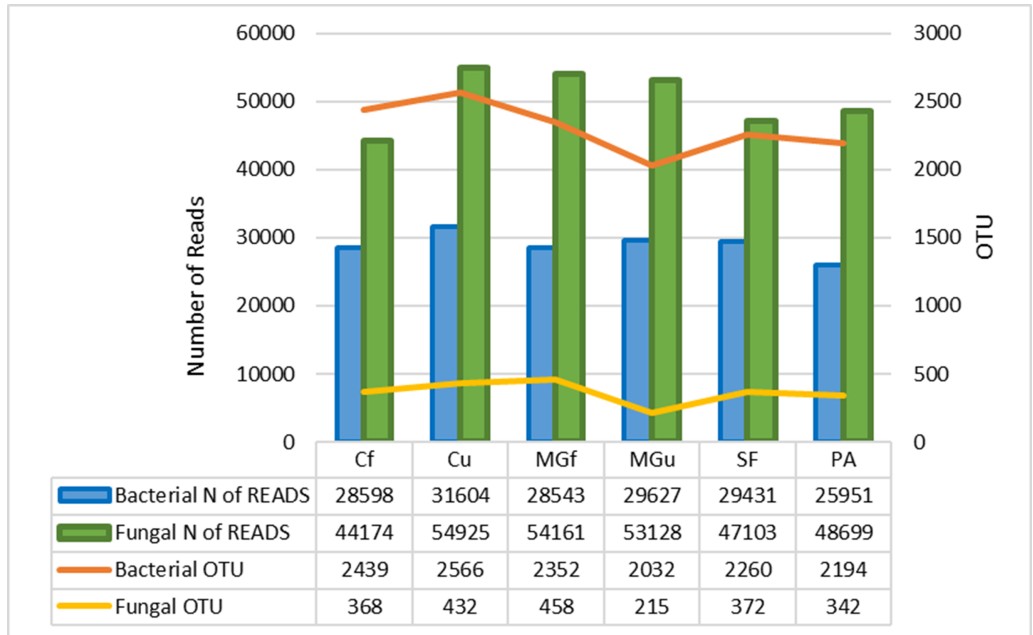

**Figure 6** Number of bacterial and fungal reads and operational taxonomic units (OTU).

clustered into 4458 smallest (at species level) OTU, were obtained, and total of 302,190 fungal rRNA spacer ITS1 valid fragments were clustered into 707 smallest (at species level) OTU (Fig. 6). On average, about 2307 bacterial taxonomic units and 365 fungal taxonomic units were determined for each sample tested. The highest number of reads for both bacteria and fungi were in the unfertilized cropland, and the OTU of bacteria was mostly in the unfertilized grassland sample, and that of fungi in the fertilized cultivated grassland (Fig. 6). Heat maps (Fig. 7) were built by applying the NG-CHM Heat Map Builder 2.20.2 (*Ryan et al., 2019*).

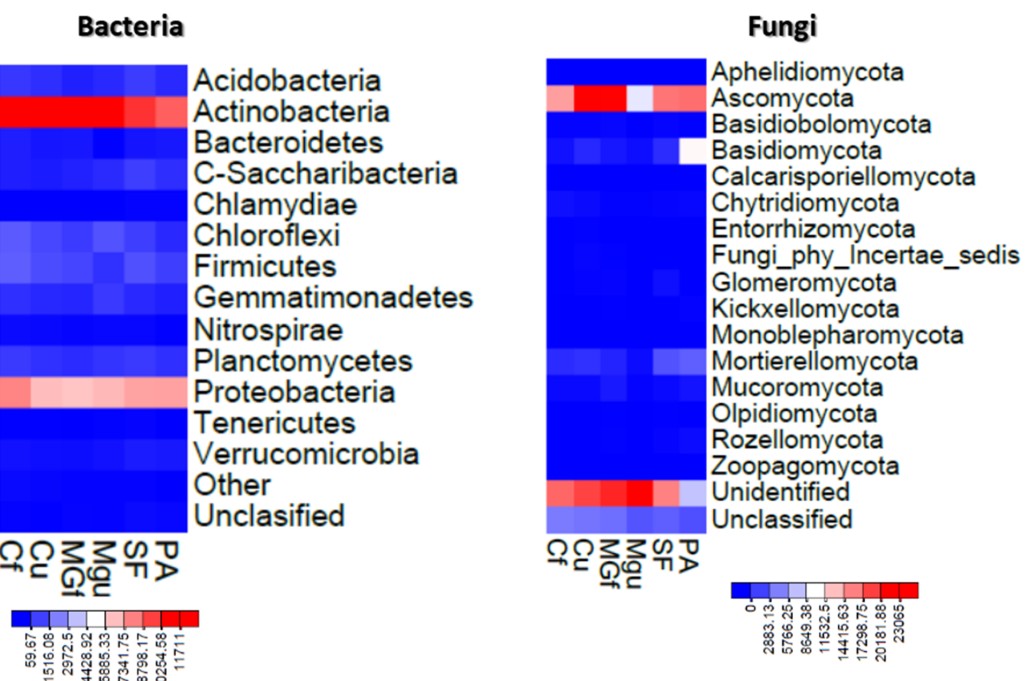

**Figure 7   Heatmaps constructed based on abundance data of bacterial and fungal OTU.** The number of reads indicated on the scale.

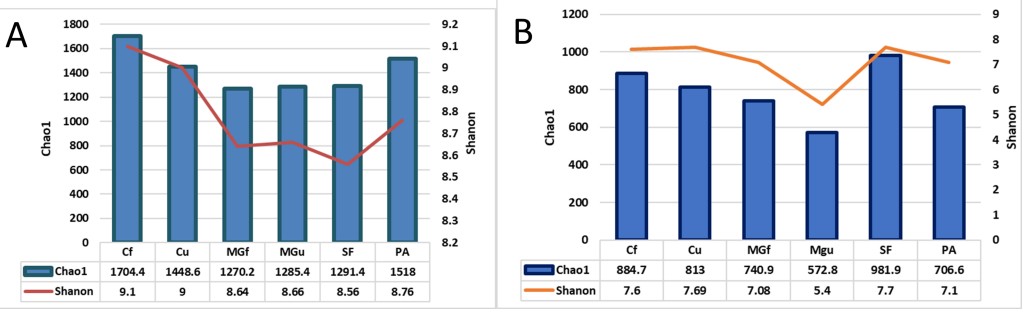

**Figure 8   The bacterial (A) and fungal (B) alpha diversity parameters in investigated samples of renaturalizated areas.**

After calculating alpha abundance parameters, it was found that the highest species richness of bacteria was in fertilized cropland (Cf) and planted plots (1704.4), while the highest species richness (Chao1) and abundance (Shannon index) of fungi were highest in soiled field (Sf) (respectively, 981.9 and 7.7). The distribution of fungi varied quite a bit, and the lowest was in the unfertilized managed grassland (MGu) plot (Chao1 = 572.8, and Shanon = 5.4) (Fig. 8).

Analyzing the taxonomic diversity of soil microbes, it was observed that the bacterial communities were dominated by two types of bacteria: *Actinobacteria* and *Proteobacteria*.
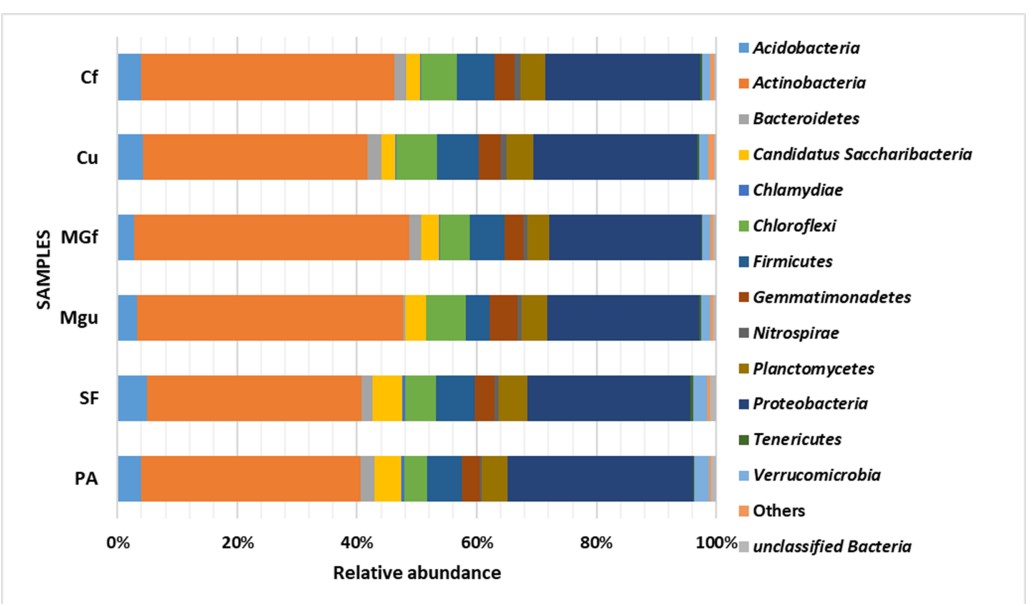

**Figure 9** Relative abundance of most common bacteria phyla.

The highest number of *Actinobacteria* was in the managed grassland (MGf) (45.88%), the least in the afforested area PA (26.68%). The amount of *Proteobacteria* varied from 25.28% in the MGf area to 30.95% in the afforested area. The amount of other bacteria important for agroecosystems belonging to the *Firmicutes* phylum varied from 3.95% in managed grassland (MGu) to 7.01% in fertilized cropland (Cf) (Fig. 9). The distribution of fungi was slightly different. The main taxonomic part of all fungi was occupied by representatives of *Ascomycota* phyllum (59.61–70.02%), but the area that was planted with pine trees stood out here, *Ascomycota* occupied only 45.28%, and 29.5% of the space was transferred to *Basidiomycota*. Meanwhile, *Basidiomycota* were few in other plots: only from 3.09 to 6.71%. Another taxonomic group of fungi accounted for an appreciable proportion was *Mortierellomycota* (3.24–12.48%) (Fig. 10).

## DISCUSSION

Changing the use of agricultural land to forestry or other land uses that the annual crop and harvest cycle will be replaced by other cycles, *e.g.*, significantly longer forest cycles. As a result, the physico-chemical properties of the soil use change, which has a decisive influence on the dynamics of nitrogen and carbon fluxes (*Jones et al., 2004*; *Li, Niu & Luo, 2012*; *Liu, Shao & Wang, 2013*). During the land use change process, aboveground vegetation changes lead to changes in the underground communities of microorganisms (*Mikkelson et al., 2013*).

Various agrochemical and botanical studies in the areas of the long-term renaturalization experiment have been carried out since 1995 (*Kazlauskaite-Jadzeviče et al., 2019*; *Tripolskaja et al., 2022*). If the agrochemical indicators were checked periodically from the beginning of the experiment, then the microbiological analysis was conducted only in 2017–2020.

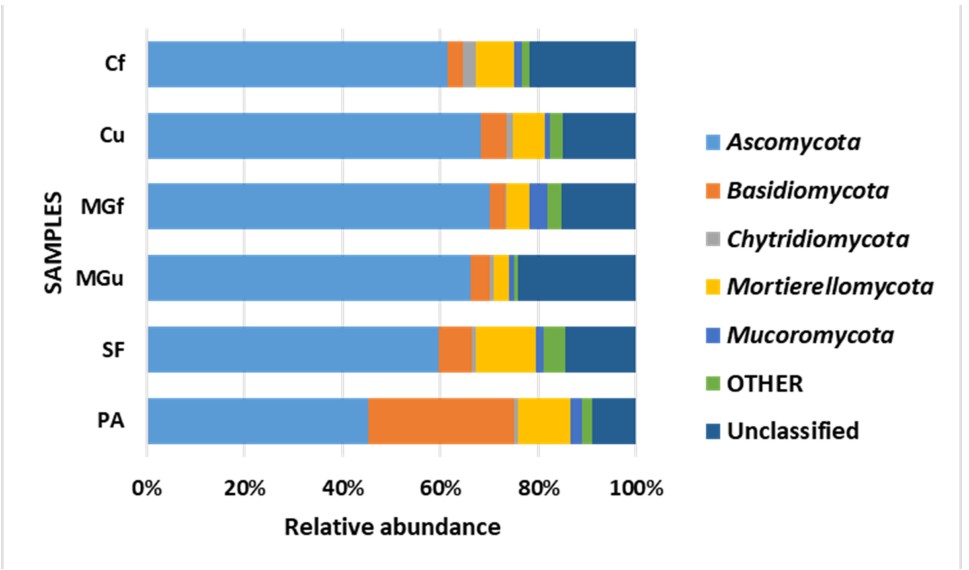

**Figure 10 Relative abundance of most common fungal OTU under the phylum level.**

Therefore, it is only necessary to compare the data of these studies among individual areas during the present period of investigation. As the research years were characterized by very different meteorological conditions (Fig. 2), it would be impossible to determine some trends in the dynamics of the microorganism's abundance. Therefore, we will analyze the data for each year separately (Figs. 3, 4 and 5).

Summarizing the dynamics of bacterial abundance during the study period, it should be noted that the afforested area differed in the smallest amounts and fluctuations in abundance. Meanwhile, in terms of fungi and yeasts, this area was characterized by higher volumes. Such a tendency is confirmed by studies by other authors (*Carson et al., 2010*; *Ren et al., 2016*; *Dang et al., 2018*; *Wang et al., 2019*).

Comparing the results of the analysis of cultivable bacterial abundance with each other, we notice that the largest fluctuations were in the areas where some anthropogenic activity was carried out. Particularly sharp jumps in the abundance of diazotrophs and nitrifiers were caused by fertilization and the cultivation of certain plants (leguminous). As organic fertilizers were not used in the studied areas, organotrophic bacteria were not so abundant (Figs. 3 and 5). In some cases, such as in 2020, the amount of organotrophs in summer samples was significantly lower than at the beginning or end of vegetation (Fig. 5). This was most likely due to a sufficiently high temperature and lack of humidity. The highest abundance of organotrophic bacteria was recorded in the summer–autumn of 2019 ($2.15 \pm 0.06$ and $2.39 \pm 0.03 \times 10^5$ CFU $\times$ g$^{-1}$). The highest abundance of non-symbiotic diazotrophs was found in all areas undergoing anthropogenic activity in the summer of 2017 (from $2.92 \pm 0.02$ to $3.13 \pm 0.06 \times 10^5$ CFU $\times$ g$^{-1}$). In some cases, their number was higher in the cropland (fertilized and not) in autumn 2019 and spring 2020 (Fig. 5). A statistically higher abundance of nitrifiers was found in croplands in autumn 2019 ($3.67 \pm 0.67 \times 10^5$ CFU $\times$ g$^{-1}$) and spring 2020 ($3.72 \pm 0.46 \times 10^5$ CFU $\times$ g$^{-1}$). High levels of

fungi were detected in the fertilized part of the cultivated grassland in the summer–autumn period of 2019–2020 ($7.46 \pm 0.38$–$9.6 \pm 0.11 \times 10^3$ CFU $\times$ g$^{-1}$) and in the afforested area in summer–autumn 2017 ($8.7 \pm 0.12$–$7.03 \pm 0.09 \times 10^3$ CFU $\times$ g$^{-1}$) (Fig. 4). The lowest and most stable bacterial amounts were in the afforested area, which, considering the changes in agrochemical parameters during renaturalization, accumulated the highest amount of soil organic carbon up to $12.2 \pm 0.1$ g $\times$ kg$^{-1}$ and the highest in the humification rate, reaching 21.3% (*Tripolskaja et al., 2022*). These processes were significantly influenced by the higher amount and the taxonomic structure of fungi compared to other samples.

In other nearby experiments with intensive and organic farming, the cultivable bacterial abundance in the low-yield sandy loam (*Haplic Luvisols*) soil was $10^5$–$10^6$ CFU, the fungal was $10^3$–$10^5$ CFU (*Bakšiene et al., 2007*; *Sivojiene et al., 2021*), in the loamy *Cambisol* bacteria—up to $10^6$ and fungi $10^5$ (*Jurys & Feiziene, 2021*). In fertile soils carbonate Chernozem in Kazakhstan, researchers counted organotrophic and nitrifying bacteria up to $10^7$ CFU and fungi up to $10^5$ CFU (*Churkina, Kunanbayev & Akhmetova, 2012*). Thus, we see that the amounts of microorganisms can vary tens of times depending on the type of soil.

Alpha diversity indexes were calculated to assess species diversity: Chao1 and Shannon. The Chao1 index estimates the total richness. The Shannon Diversity Index is a mathematical measure of the diversity of species in each community. This index provides more information about the composition of the community, i. considers the relative abundance of different species and Chao1 the species richness in each community (*i.e.,* the number of existing species). The highest richness of bacterial species was observed in the fertilizing part of the crop rotation field, and fungi—in the soiled area (Fig. 8).

Most amount of the DNA sequences were assigned to 13 major bacterial phylum (Fig. 9). The taxonomic groups of *Actino-* and *Proteobacteria* were the most numerous. The content of *Actinobacteria* ranged from 43% in the cultivated grassland to 34% in the fallow (SF); most of the *Proteobacteria* were 29% in the afforested area, the lowest in the cultivated grassland 24% (Fig. 9). *Ren et al.* state (*2016*) that *Proteobacteria* predominate in place of former *Actinobacteria* in the soil of the afforested area. In the case of our study, this was the case compared to the cropland, the relative amount of *Proteobacteria* in the cultivated soil increased and the number of *Actinobacteria* decreased (Figs. 9 and 7). The third taxonomic category in terms of quantity in our case is *Firmicutes* (5.29%), not *Acidobacteria*, as stated by *Ren et al. (2016)*. According to the data of other authors (*Wang et al., 2019*), when the former grassland is planted, the indicators of bacterial abundance are reversed—from *Proteobacteria* to *Actinobacteria* (Figs. 9 and 7).

According to the data of fungal metagenomic analysis, all read DNA fragments were organized into five main large taxonomic categories (Figs. 10 and 7) and the remaining large category of unclassified functional units. *Ascomycota* had the largest number of taxonomic units (from 37% to 42% of all fungi). Comparing the taxonomic structure of all the studied samples, we can see that the structure of the afforested area differs. An increase (up to 24%) in the taxonomic group of *Basidiomycota* is observed here at the expense of *Ascomycota* (Figs. 10 and 7). In all remaining samples, *Mortierellomycota* was second in abundance (range 1% in unfertilized grassland to 7.92% in fallow). The

most common members of this taxonomic group were the genus *Mortierella*. The increase in *Basidiomycota* in the pinus planted area is not surprising, as the pine root system is characterized by mycorrhiza and most mycorrhizal fungi belong to *Basidiomycota*. Other researchers have confirmed this statement in their work (*Dang et al., 2018*; *Wang et al., 2019*; *Xue et al., 2022*). Representatives of the following genera of fungi have appeared in the afforested area: *Inocybe*, *Russula*, *Tomentella*, *Pseudotomentella*, *Tricholoma*, *Tylospora* and others.

Larger substantial changes are observed by analyzing the structure of the lower taxonomic ranks of *Ascomycota*. The most prominent of all is the samples from non-anthropogenic fields, i. fallow and afforested areas. The number of taxonomic units belonging to the orders *Eurotiales* (genera *Penicillium*, *Aspergillus*, *Talaromyces*) and *Hypocreales* (*Acremonium*, *Metarhizium*, *Lecanicillium*, *Trichoderma*, *Fusarium*) was significantly higher in the pine planted and fallow. This was especially evident in the sample of the afforested field, the increase of the representatives of these orders is at the expense of the order of *Pleosporales* (genera *Coniothyrium*, *Pyrenochaeta*, *Pleotrichocladium*), which is more numerous in the cropland. In the afforested area there are several representatives of the order *Helotiales* (*Meliniomyces* (mycorhyzal fungi), *Tetracladium*, *Cadophora* (mycorhyzal fungi), *Phialocephala* (mycorhyzal fungi). In the area planted with pines, there was an increase in the taxonomic rank of *Basidiomycota* fungi, including many mycorrhizal fungi belonging to the genera *Inocybe*, *Tricholoma*, *Tylospora*, *Russula*, *Pseudotomentella*, *Tomentella*, *Naganishia*). A distinctive feature of the afforested area was the appearing of a basidiomycetous yeasts at the species level, *Slooffia cresolica*, covering 2429 reads.

Thus, the form of renaturalization-afforestation had the greatest impact on soil microorganism communities. The largest structural changes occurred here, especially among fungi (Figs. 10 and 7). However, it is this area that has suffered from some pests, which has destroyed almost all the trees in the last few years. Therefore, when planning future afforestation, the phytosanitary condition needs to be closely monitored and the necessary measures taken in a timely manner.

## CONCLUSIONS

The analysis of the abundance of bacteria and microscopic fungi showed that their abundance depends on the applied agrotechnical measures and the specifics of the cultivated plants, as well as on the meteorological conditions. The abundance of both cultivable bacteria and fungi was not high compared to other types of soils, bacteria were counted up to $10^5$, and fungi—$10^3$ CFU per 1 g of dry soil. In the renaturalized areas, where no economic activity took place, the abundance of microorganisms was statistically lower and less variable in terms of abundance during the vegetation period than in the cultivated land areas. Summarizing the dynamics of bacterial abundance during the study period, it should be noted that the area planted with pines differed in the smallest amounts and fluctuations in abundance. In the case of fungi and yeasts, meanwhile, the area was more abundant. The taxonomic groups of *Actinobacteria* and *Proteobacteria* had the highest OTU. The relative amount of *Proteobacteria* increased and the number of *Actinobacteria*

decreased in the area planted with pines compared to other. The highest number of fungal OTU is characterized by the division of *Ascomycota*. From all the studied samples, the taxonomic structure differs from the afforested area, which, at the expense of *Ascomycota*, significantly increased the number of *Basidiomycota* (especially mycorrhizal). To maintain a stable structure of soil microorganisms' communities, moderate fertilization with both mineral and organic fertilizers should be applied, as well as a fair crop rotation, especially for bean crops. The choice of afforestation requires regular monitoring of the phytosanitary status and preventive measures against diseases and pests and timely protection measures. Further studies should try to determine what period is needed for the reorganization of soil microcommunities from the initial phase up to the present.

### Funding
The authors received no funding for this work.

### Competing Interests
The authors declare there are no competing interests.

### Author Contributions
- Audrius Kacergius conceived and designed the experiments, performed the experiments, analyzed the data, prepared figures and/or tables, authored or reviewed drafts of the article, and approved the final draft.
- Diana Sivojiene performed the experiments, analyzed the data, authored or reviewed drafts of the article, and approved the final draft.

### Data Availability
  The data is available at the Sequence Read Archive (SRA): Bioproject PRJNA842894; SAMN28798056 to SAMN28798065.

### Supplemental Information
Supplemental information for this article can be found online at http://dx.doi.org/10.7717/peerj.14761#supplemental-information.

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
