# Peer review of "Microbial diversity and abundance in loamy sandy soil under renaturalization of former arable land"

_PeerJ, doi:10.7717/peerj.14761_

## Round 0.1 · original submission · Major Revisions

· Academic Editor

Major Revisions

Dear authors, your contribution has been evaluated by experts in the field and their comments for incorporation have been enclosed. Additionally, authors need to focus more on the Methodology section by describing the protocols in an explicit style. Moreover, the study rationale must be established in the Introduction section. Last but least, authors must consider modifying the Results and Discussion section by highlighting pertinent findings, interpreting those along with correlating them with latest peer-findings.

Reviewer 1 ·

Basic reporting

The manuscript submitted by Kacergius et al. entitled “Microbial diversity and abundance in loamy sandy soil under renaturalization of ex-arable land” evaluated soil microbial populace (bacteria and fungi and yeasts) in soils having conversion of arable land to cultivated meadow, soil and forest, leaving the experimental area of arable land. The content of the contribution might be of interest to the readers of PeerJ, however authors must consider clarifying below points for reader’s convenience.
ABSTRACT
The starting phrase need to identify the problem statement or study rationale in order to establish study justification but it is lacking rather a too generalized statement has been added that must be replaced.
Results description must follow methodology but statements like “The taxonomic structure was determined in 2020 by Next Generation Sequencing of summer soil samples” in results may confuse the readers.
In order to impart briefness, following sentences must be merges into one;
The taxonomic groups of Actino- and Proteobacteria had the highest bacterial abundance. The content of Actinobacteria ranged from 34% in the soiled field to 43% in the grassland; Proteobacteria in the grassland at least 23% and in the afforested area up to 29%. The highest number of fungal OTU was distinguished by Ascomycota fungi (37–42% of the total number of fungi).
INTRODUCTION
In order to avoid the impression of a localized study, it is perhaps better to enrich introduction with global information on renaturalization and their consequences.
Additionally, peer-findings on the subject matter need to be described in a critical manner by highlighting research and knowledge gaps.
Ex-arable terminology needs to be replaced with an appropriate term.
More information on alpha diversity index may be added.

Experimental design

Climatic conditions need to be briefly described by omitting unnecessary details.
Critical information regarding methodologies/protocols used for identification of microbial communities need to be described in detail.
It is perhaps better to present protocols and equipment used during course of the study in tabular form.
Heat maps preparation information may be removed from statistical analyses.
More importantly, after reading the entire methodology, readers may remain clueless about employed treatments and response variables of the study, thus authors must consider adding these clearly.

Validity of the findings

RESULTS AND DISCUSSION
This section is the most pertinent one however it has been synthesized poorly. Authors should describe results first and thereafter interpretation of recorded finding must be added. It should following correlating recorded results with peer-findings step by step in accordance with recorded data.
CONCLUSIONS
It lacks future perspectives of the study along with limitations of the study.

·

Basic reporting

Article is written technically correct. Article is having sufficient introduction and background to demonstrate the work. Relevant review of literature is provided. Relevant data is provided in the form of Figures .

Experimental design

All the experiments are conducted to a high technical standards. All methods are described in a well mannered way.

Validity of the findings

The data provided is statistically sound.

Additional comments

The manuscript titled “Microbial diversity and abundance in loamy sandy soil under renaturalization of ex-arable land” is written quiet well. It can be accepted after the incorporation of few suggestions. Some of my suggestions regarding the manuscript are
• Abstract is written quiet well.
• Keywords are missing. Provide these ones.
• Introduction part is written quiet well. It would be much better too keep the important text in the introduction and remove the extra text. Introduction is too much, it’s much better to reduce that part. Also there are too many references. Keep the latest ones and old references should be removed.
• Materials and methods part is written quiet well.
• Results and discussion part is also written well. It would be much better if authors could write the results and discussion part separately for better understanding.
• Try to improve the quality of all figures.
• If possible add the Standard Error in all figures.
• References should be according to the journal format.

Reviewer 3 ·

Basic reporting

In this study clear language is used throughout, literature is well referenced and relevant. Figures are relevant.
The manuscript presents original research which is within a scope of the journal. Research question is well defined and relevant. The incestigation was performed to well-known standards.

Experimental design

Methods should be described better as it is not clearly stated how many soil samples were NGS sequenced. What bioinformatic tools were used? The authors did not provide link to the raw NGS data.

Validity of the findings

No comment

Additional comments

I believe the manuscript needs to be improved and the methodology part should be described in more details.

---

## Round 0.2 · Minor Revisions

· Academic Editor

Minor Revisions

Dear authors, one reviewer in the field has acknowledged improvements made in your contribution, however, there are still some serious shortcomings.

In the abstract, a problem statement at the beginning is still missing. Additionally, confusion exists regarding the treatments and response variables as pointed out in the previous review round.

The introduction section still contains too many generalized statements while authors need further to enrich it by highlighting research gaps. The most serious concern is a results section that is too short, while the discussion section must be made a bit briefer while focussing on the interpretation of recorded data by avoiding too many speculations.

·

Basic reporting

The article is written in English and clear.
Sufficient introduction and background knowledge is provided.
Figures and tables are also quiet clear.

Experimental design

Methods are described in a comprehensive way.

Validity of the findings

Results finding are quiet clear.

Additional comments

Overall the authors have made the revision in a quiet good way. So it can be accepted as such.

---

## Round 0.3 · Minor Revisions

· Academic Editor

Minor Revisions

Dear Authors, although reviewers comments have been sufficiently addressed but still there are few critical deficiencies that have been highlighted in the attached file. Those suggestion must be incorporated in next version of your contribution.

---

## Round 0.4 · accepted · Accept

· Academic Editor

Accept

Dear authors, On the recommendation of reviewers and looking into the incorporations made by authors, I am satisfied with the improvements and pleased to convey that the manuscript is accepted for publication in PeerJ.